# Neuroplastic Changes in Addiction Memory—How Music Therapy and Music-Based Intervention May Reduce Craving: A Narrative Review

**DOI:** 10.3390/brainsci13020259

**Published:** 2023-02-03

**Authors:** Filippo Pasqualitto, Francesca Panin, Clemens Maidhof, Naomi Thompson, Jörg Fachner

**Affiliations:** 1Cambridge Institute for Music Therapy Research, Anglia Ruskin University, Cambridge CB1 1PT, UK; 2School of Life Sciences, Faculty of Science and Engineering, Anglia Ruskin University, Cambridge CB1 1PT, UK

**Keywords:** music therapy, music-based interventions, perineuronal nets, craving, addiction memory, substance use disorder, neuroplasticity

## Abstract

Recent findings indicate that Music Therapy (MT) and Music-Based Interventions (MBIs) may reduce craving symptoms in people with Substance Use Disorders (SUD). However, MT/MBIs can lead SUD clients to recall memories associated with their drug history and the corresponding strong emotions (addiction memories). Craving is a central component of SUD, possibly linked to relapse and triggered by several factors such as the recall of memories associated with the drug experience. Therefore, to address the topic of what elements can account for an improvement in craving symptoms after MT/MBIs, we conducted a narrative review that (1) describes the brain correlates of emotionally salient autobiographical memories evoked by music, (2) outlines neuroimaging and neurophysiological studies suggesting how the experience of craving may encompass the recall of emotionally filled moments, and (3) points out the role of perineuronal nets (PNNs) in addiction memory neuroplasticity. We highlight how autobiographical memory retrieval, music-evoked autobiographical memories, and craving share similar neural activations with PNNs which represent a causal element in the processing of addiction memory. We finally conclude by considering how the neuroplastic characteristics of addiction memory might represent the ground to update and/or recalibrate, within the therapy, the emotional content related to the recall.

## 1. Introduction

It is estimated that substance use is directly (i.e., substance overdoses) and indirectly (i.e., substance use as a risk factor for premature death) responsible for 11.8 million deaths each year [1]. The Diagnostic and Statistical Manual of Mental Disorders (DSM-5) defines Substance Use Disorder (SUD) as a “cluster of cognitive, behavioral, and physiological symptoms indicating that the individual continues using the substance despite significant substance-related problems” [2] (p. 483). SUD is a multidimensional condition [3,4] that has been described as a chronic and relapsing disorder [5,6,7]. SUD, and mental disorders in general, are difficult to frame into a single definition or a mere label [8], and diagnostic terminology can be stigmatizing and perceived as objectifying and insulting [9]. Notwithstanding, nosological definitions and criteria are still useful, when pursued sensitively and with adequate training, to set a threshold for what can be considered a “disorder” and to convey relevant information in treatment settings [8,9]. A key factor for SUD is the experience of craving, which is defined as the memory of drugs’ rewarding effects superimposed upon a negative emotional state [10,11] and manifests itself as an “intense desire or urge for the drug” [2] (p. 483).

Memory is one of the most relevant cognitive functions affected by addiction and involved in drug cravings [12,13]. A pathological memory related to addictive behavior has been named addiction memory [12,13]. We use this expression, throughout this article, to refer to a composite memory related to drug dependence [13]. Addiction memories can be state-dependent, i.e., memories related to how the drug makes a person feel and/or drug-related, i.e., memories of drug-related cues and/or contexts [13]. Drug-related cues are discrete elements of the environment that a person associates with a previous drug experience (e.g., a glass pipe, a needle, a pack of cigarettes), and a drug-related context represents the general environment (e.g., a social setting) that the person associates with a previous drug experience [13,14]. These memories exhibit a neuroplastic feature, indicating that they are not static and fixed. Indeed, they are flexible and dynamic due to cognitive strategies (see Section 4) and neurobiological mechanisms (see Section 4 and Section 4.2). Because of addiction memory neuroplasticity, interventions have a chance of attenuating such memories and possibly stifling the loop of drug relapse [13].

The historical link between music and drugs is related to the ability of both to alter our emotions [15], our memories [15,16,17], and our state of consciousness [16]. As a form of communication and as a therapeutic and relational tool to improve health and well-being, music has long been a significant environmental stimulus for humans [18,19]. Music Therapy (MT) is defined as a clinical intervention delivered by an accredited music therapist which adopts music as a therapeutic tool to accomplish individualized goals within a safe therapeutic relationship [19,20]. In the UK, music therapists register with the Health and Care Professions Council on completion of a 2-year full-time or 3-year part-time masters level training. Furthermore, for the purposes of this narrative review, it was worth including other experimental protocols investigating the therapeutic effects of music and not necessarily involving the participation of an accredited music therapist [21]. In accordance with previously published work, these will be referred to as Music-Based Interventions (MBIs) [21,22].

Although recent findings of a Cochrane review [23] indicate that MT and MBIs reduce craving symptoms in SUD clients, there is still a lack of information about possible neural mechanisms of this therapeutic change [23]. Therefore, building upon the evidence showing a noticeable overlap in the activations of brain structures devoted to craving, autobiographical memory retrieval, and music-evoked autobiographical memories, the aim of this narrative review is to answer the following question: What factors can explain the reduction in craving symptoms after MT and MBIs? Or to be more specific: Can the neuroplasticity of addiction memory represent viable access for MT and MBIs to reduce craving in SUD participants?

To address this question, we conceived a three-folded purpose developed into three different sections. The first objective (Section 2) is to describe brain correlates of how music can cue autobiographical memories and evoke emotions. The second objective (Section 3) is to emphasize that the feeling of craving is associated with neurophysiological and metabolic activity in regions of the brain thought to play a role in emotional processing and autobiographical memories, suggesting how the experience of craving may entail the recall of emotionally filled moments. To do this, we describe how this construct has been addressed in the human neuroscience field and we outline the overlap between brain regions involved in craving, autobiographical memory retrieval, and music-evoked autobiographical memories (MEAMs). The third and last objective (Section 4) is to describe the role of specific molecular aggregates called Perineuronal nets (PNNs)—specialized extracellular matrix structures surrounding the neurons and involved in synaptic plasticity [24]—on the acquisition and reconsolidation of addiction memories. Here, we highlight PNNs’ presence in brain regions involved in emotional processing, autobiographical memory, and the feeling of craving in humans, emphasizing their role as a possible molecular system that addiction treatments can target to reduce craving. We finally provide a conclusion (Section 5) that emphasizes the similarities in the neural underpinnings of MEAMs, music-evoked emotions, the feeling of craving in humans, and the neuroplasticity of addiction memories in animals, and we propose a new perspective on how MT and MBIs may reduce craving in SUD, i.e., the role of PNNs for emotion and memory in the reduction of craving with MT/MBIs.

### Methods

We conducted a narrative overview [25] to summarize findings from different research fields supporting a new idea that has not been discussed yet in the field of MT/MBIs for SUD. We aimed to review articles in the neuroscience of music and emotion, music-evoked autobiographical memories, the retrieval of autobiographical memories, the craving state, and clinical articles on the effect of MT/MBIs for SUD. The studies examined here adopted different methodologies and collected data from different samples. Indeed, we reviewed studies employing electroencephalography (EEG), functional magnetic resonance imaging (fMRI), positron emission tomography (PET), qualitative measurements (such as interviews or open-ended questions), and quantitative subjective measurements (such as multi- or single-item questionnaires) in human research models, and preclinical research models. The heterogeneity of the sources included and the novelty of some research fields (e.g., there is still no research that adopts neuroscience techniques in the study of MT/MBI for SUD) determined the choice to conduct a narrative review. While considering the limitations of adopting this approach (see Section 5.2 “Strengths and Limitations”), this choice allowed us to incorporate different methodological and theoretical frameworks which is an essential component to address a novel question in the interdisciplinary field of MT for SUD.

We conducted targeted literature searches for our fields of interest on the databases “PubMed” and “Cochrane Database of Systematic Reviews” as well as on the search engine “Google Scholar”. Search terms were: (“Music Therapy” or “Music-Based Interventions”) and (“Substance Use Disorder”) and (“craving”) and (“memory” or “addiction memory” or “autobiographical memory” or “music-evoked autobiographical memory”) and (“emotions” or “music-evoked emotions”) and (“EEG” or “fMRI” or “PET” or “preclinical study”) and (“neuroplasticity” or “brain plasticity”). For the purposes of the review and to maintain coherent and comprehensible writing along with the analysis of different research fields, we decided not to critically analyze each study included in depth, remaining within the methodological framework of narrative reviews [25,26].

## 2. Music’s Ability to Elicit Emotions and Evoke Memories: A Look at the Brain Correlates

In the last two decades, MT and MBIs have been increasingly recognized as useful options for the treatment of several complex conditions, such as stroke, schizophrenia, dementia, autism spectrum disorder, anxiety, depression, and Parkinson’s disease [27,28,29,30,31,32,33], and neuroscientific investigations concerning aspects of MT and MBIs are becoming more and more frequent [34]. Musical interactions in a group or individual therapeutic setting could include, but are not limited to, free improvisation, structured musical activities, playing pre-composed songs as chosen by the client, and listening to pre-recorded music. Furthermore, within the safety of the therapeutic relationship, unconscious processes taking place in the pre-verbal musical communication may be able to be verbalized [35].

### 2.1. Neural Correlates of Music-Evoked Emotions

The use of neuroscience methods in MT/MBIs settings with clinical and non-clinical participants is a powerful means to uncover the therapeutic potential of music. One area that has been extensively explored concerns the emotional effects of music (music-evoked emotions). Emotional responses to music involving a complex set of psycho-physiological mechanisms [36] and neuroimaging research—utilizing fMRI and PET—has shown that music appreciation is processed in similar brain regions as other highly rewarding and emotional stimuli, such as psychoactive drugs [37,38,39]. Indeed, pleasurable music eliciting “shivers down the spine” or “chills” is associated with increased activity in the dorsal (caudate and putamen) and ventral (Nucleus Accumbens—NAc) striatum, ventral tegmental area (VTA), orbitofrontal cortex (OFC), anterior cingulate cortex (ACC), insula, and decreased activity in the amygdala, hippocampus, and ventromedial prefrontal cortex (vmPFC) [40,41,42,43].

Increases and decreases in the metabolic activity of these brain regions during pleasurable music listening are compatible with the role that these structures play in processing emotion and reward [37]. Furthermore, the causal role that this complex set of brain structures plays in music-evoked emotional processing has been causally demonstrated by neuropsychological studies of brain lesions [38], and a recent neuropharmacological study with healthy participants described the causal role played by the neurotransmitter dopamine in the rewarding experience induced by music [44]. The characteristic of music to induce an emotional reaction opens up various possibilities for clinical application. The fact that music-evoked emotions are processed in brain regions involved in drug craving (see Section 3) led authors and practitioners to think that working with music may help to reduce the intensity of craving in SUD individuals [16,45,46].

### 2.2. Music Therapy and Music-Based Interventions for Substance Use Disorder

Three recent systematic reviews investigated the efficacy of MT and MBIs on SUD [22,23,47] indicating beneficial effects on several SUD-related outcomes such as depressive symptoms [48,49,50,51], anxiety symptoms [50,52,53], negative emotions (i.e., anger) [50], and subjective feelings of craving [54,55,56]. Reductions in craving symptoms have been identified as a main outcome of a Cochrane systematic review when participants received MT in addition to standard care, as compared to participants receiving standard care alone [23]. However, despite empirical evidence suggesting that MT/MBIs are effective for SUD individuals, results are not consistently reported across studies that implement heterogeneous methodologies and types of MT/MBIs intervention [22]. Likewise, despite reported reductions in participants’ feeling of craving, the mechanism of therapeutic change underlying the effects of MT/MBIs for craving is yet to be identified [23], and future studies, such as randomized control trials adopting neuroscience research methods, may help to do so [46]. On the other hand, it has been suggested that music itself may represent a drug-related cue [45,57,58] and, by triggering state-dependent memories and inducing strong emotions, can possibly lead to a craving state [16,45,59].

In a recent systematic review about the possibility of music to induce craving, the authors concluded that (1) music has the potential to induce alcohol, cannabis, nicotine, and general substance craving; (2) participants are mainly adults living in community centers and their level of craving is generally assessed using subjective measures: Likert-type scales or multi-item questionnaires; (3) most of the reviewed studies used music as a mood-induction stimulus or in virtual reality settings, and accurate details about the characteristics of music are often neglected; and (4) most of the studies that were examined met the inclusion criteria for the systematic review on music-induced craving but not the checklist for properly reporting music-related characteristics when music is utilized in therapeutic contexts [60]. Overall, these findings suggest that music might be a stimulus that induces craving and that, when used in therapeutic contexts (such as MT/MBIs), might reduce craving symptoms.

The fact that a stimulus of the environment can induce craving is not necessarily a negative characteristic. Indeed, researchers and interventionists can deliberately induce cravings as a part of the treatment program, as happens in cue-exposure treatments [61,62] and systematic desensitization [63,64]. Notwithstanding, in the event that the client feels a music-induced craving, the therapist should ensure that the client is supported within the safety of the session [16,45] through recalibration and retraining of emotional responses to music with the aim to avoid relapse [16]. Finally, the question that arises and that has not been addressed by existing scientific reviews is: what elements might explain craving-related therapeutic change associated with MT/MBIs for SUD?

### 2.3. Music Is Able to Cue Emotionally Salient Autobiographical Memories

#### 2.3.1. The Role of Emotion in Autobiographical Memory Retrieval

The ability of music to evoke strong emotions is mediated by increased or decreased metabolic activity in brain structures playing a role in autobiographical memory retrieval [65,66,67,68,69]. The hippocampus, the amygdala, and the PFC (particularly the OFC) are central brain regions in this process as they have been associated with emotional processing as well as autobiographical memory retrieval [65,66,67,68,69,70]. An autobiographical memory is a type of explicit memory that has been defined as a “memory for the events of one’s life” [71] (p. 103). This type of memory can be divided into two dissociable sub-components: the recall of personal semantic information and personal episodic information. Personal semantic memory is factual and related to the self (e.g., knowing where one was born) [72] while episodic memory is about personal events requiring a relational component that bind the information to the specific context [73].

Although they have been associated with the activity of a common neural network of limbic and cortical areas, episodic and semantic memories seem to rely on different patterns of hippocampal–cortical connections and different electrophysiological correlates [72,74]. However, the dissociation of these two sub-components is a non-trivial process because when adopting certain operational definitions, the neural correlates of personal semantic memory appear remarkably similar to the ones of episodic memory [75]. Extensive research on the functional neuroimaging of autobiographical memory used PET and fMRI technologies to suggest that this is not stored within a single area of the brain but seems to be the result of a distributed network throughout the brain [65,66,67,68,69,70].

Therefore, an autobiographical memory network including “search and controlled retrieval processes”, “self-referential processes”, “recollection”, “emotional processing”, “visual imagery”, and “feeling-of-rightness/monitoring” has been conceptualized [67]. “Search and controlled retrieval processes” have been localized in the dorsolateral prefrontal cortex (dlPFC) and ventrolateral prefrontal cortex (vlPFC) [67,68,69]. Those processes characterize a voluntary autobiographical memory retrieval where the active search is guided by a semantic knowledge about the world and the self, and the control is conceptualized as a probabilistic process (“e.g., you probably went with Claire, who loves Chinese food”) [67] (p. 2019). The “feeling-of-rightness” is a pre-conscious form of monitoring that has been associated with the activity of the vmPFC [66,67,76]. “Self-referential processes” concerns the activation (less deactivated) of the medial prefrontal (mPFC), and the posterior cingulate (PCC) cortices that occur when participants recognize familiar experimental stimuli as opposed to non-familiar ones [66,67,69,77]. “Emotional processing” refers to the emotional content and the vivid sensory details that constitute the recollection of a memory; the emotional component of autobiographical memory has been associated with the activity of the amygdala, the hippocampus, and the OFC [65,66,67,68,69,70]. Interestingly, autobiographical memory retrieval has been associated with increased amygdala-hippocampus connections relative to semantic retrieval [78]. This has been interpreted as a mechanism shaped by evolution whereby the memory system preferentially retains information relevant for survival, and thus, is filled with strong emotions and motivational goals. Consequently, the effects of emotional processing on autobiographical memory have been studied and the authors suggest that emotional arousal/emotional processing improves the recollection of autobiographical memories with associated contextual elements (e.g., time, location, and sensory details) to a greater degree than the recall of a past event without associated information (e.g., having a feeling of familiarity without recalling specific details—the “when” and “where”) [67,68,69]. Finally, the bilateral visual cortex, cuneus/precuneus regions have been related to the processing of visuospatial imagery of autobiographical memory retrieval [66,67,69]. Moreover, from a cognitive-systems perspective, the recall of personal episodic information does not occur from a general cognitive structure about unified information but rather from the integration of different basic, low-level systems relying on partially different brain substrates [79,80]. For instance, recalling a relevant episode of one’s life may involve language, vision, audition, olfaction, spatial imagery, and emotion [79,80]. In a sense, remembering information about relevant events in one’s life means reliving some aspects of that moment. Emotion is part of multiple basic low-level systems involved in the process of recalling an event because we do not recollect any kind of memory in the same way and with the same level of vividness. Moments imbued with emotion are generally recalled the most vividly and durably [67,81]. To re-construct an episodic autobiographical event we need to encode the event, consolidate it into a stable memory representation, and retrieve it successfully. Emotions can influence each one of these passages [81].

#### 2.3.2. Neural Correlates of Music-Evoked Autobiographical Memories

Neuroimaging fMRI technology has been used to probe the functional brain activations of music-induced autobiographical memories (MEAMs) [82,83,84]. MEAMs are arrays of memories elicited by music and displaying common features. They are most likely involuntarily recalled from within the reminiscence bump period [85] which is an increased number of memories older people recall from 10 to 30 years of age [80]. Moreover, MEAMs are mainly relative to significant people (social theme), imbued with an emotional reaction (mainly positive—happy and youthful memories; and nostalgic) [86] and vivid, being described with a higher prevalence of perceptual and internal details compared to memories elicited by famous faces [87]. Perceptual details reflect sensory experiences such as sights, sounds, and smells while internal details have been categorized as including “events (e.g., happenings, actions taken), places (e.g., room, building, city), times (e.g., year, month, semester), perceptions (e.g., smells, sights, sounds), and emotions/thoughts (e.g., happiness, sadness) directly related to the memory” [87] (p. 4).

An fMRI study showed increased metabolic activity of the dlPFC and the vlPFC positively correlating with ratings of musical stimuli as autobiographically salient [84]. This finding led the author to the interpretation that the mPFC is a critical processing hub for MEAMs. In support of this interpretation, a study [82] used fMRI to measure the brain activity of people retrieving autobiographical memories while listening to familiar and popular songs compared to non-familiar popular songs. MEAMs have been associated with increased activity in the mPFC (both vmPFC and dorsomedial prefrontal cortex—dmPFC), dlPFC, vlPFC, OFC, hippocampus, amygdala, and posterior cingulate cortex (PCC) [82].

While the above-mentioned studies [82,84] focused on a sample of young healthy adults, a different study investigated the neural correlates and the phenomenological differences of MEAMs in older and younger healthy adults [83]. This fMRI research confirmed the brain activity data of previous studies and pointed out age differences in the phenomenological characteristics of MEAMs [83]. Indeed, young adults displayed an enhancement of memory details compared to older adults, who, in contrast, showed a greater affective positivity effect (i.e., more positive autobiographical memories). The authors also suggest that mnemonic enhancement as a function of song familiarity is associated with increased recruitment of dmPFC in older adults and vmPFC and hippocampus in younger adults [83].

In the previous sections, we have addressed the ability of music to evoke emotions (Section 2.1), the role of emotion in the process of recalling an event (Section 2.3.1), the neural correlates of MEAMs, and the strong link between emotional characteristics of the event and the memory. This suggests how music may represent a salient memory cue and reliably evoke memories imbued with emotional content. The understanding of therapeutic change as well as the differentiation of therapeutic approaches by adapting them to the variety of people’s characteristics can be benefited by furthering the study of the close relationship between MEAMs, music-evoked emotions, and, as we will address in the following sections, craving and addiction memories.

## 3. Neural Circuitry of Drug Craving—The Role of a Brain Memory Network

Substance Use Disorder (SUD) is characterized by an underlying change in brain function observable at a molecular, cellular, and systemic (i.e., circuitry) level. These changes may persist even beyond detoxification and reflect behavioral/subjective effects such as the intense urge to get the substance and the possibility of relapse [5,6]. The intense urge towards the substance has been operationalized as a feeling of craving. It is related to the cessation or the reduction of the substance and has been associated with psycho-physiological effects [2]. Physical dependence is generally reflected in withdrawal syndromes like alcohol-induced delirium tremens (DTs) and opiate-induced cold turkey [88]. These are physical states with shakes, palpitations, sweating, over-breathing, hyperthermia, raised blood pressure, and other problems up to convulsions and death [2]. Withdrawal symptoms can elicit the urge to consume the drug even though not all drugs have been associated with these symptoms. For instance, hallucinogens, cocaine, and amphetamines do not cause physical withdrawal symptoms [88]. The cessation or reduction of stimulants (cocaine and amphetamine) have been associated with dysphoric mood, fatigue, unpleasant dreams, sleep disorders, increased appetite, and psychomotor agitation/retardation causing significant daily life distress [2]. The cessation or reduction of hallucinogens may cause common visual and—less commonly—perceptual disturbances, such as geometric hallucinations, perception of movements in the peripheral visual field, flashes of colors, etc. [2]. When individuals refrain from consuming the substance, negative psycho-physiological symptoms and the feeling of craving might be experienced. Craving is considered the hallmark of SUD [89] and has been related to many drugs, such as cocaine, amphetamines, nicotine, alcohol, and opiates. Lysergic acid diethylamide (LSD), phencyclidine, benzodiazepines, antidepressants, and antipsychotics seem not to produce this feeling [88]. In this section, we describe the brain representation of the feeling of craving by summarizing evidence from human neuroscience research. The feeling of craving is associated with neurophysiological and metabolic activity in regions of the brain thought to be involved in emotional processing and autobiographical memories.

### 3.1. Evidence from Human Imaging and Resting-State EEG

Craving is a complex and multi-dimensional construct [3,4,90], difficult to measure, and not always related to a relapsing outcome [91,92]. Traditionally, the problem of measuring craving in humans has been approached by utilizing task-based cue-reactivity neuroimaging studies and resting-state EEG. The former requires the participant to be exposed to drug-related stimuli and proper control stimuli to observe peculiar brain metabolic activity given by the difference between these conditions [93,94,95,96,97,98,99,100,101,102,103,104,105,106,107,108]. The latter involves a continuous recording of the EEG activity for 3 to 5 to 10 min while the participant sits with eyes opened and/or closed after a varying period of abstinence (from 24 h to 7 days, to 90 days, to 6 months) [99,109,110,111].

A classical PET study [97] showed the activation of a memory system of the brain during cue-elicited cocaine craving. Long-term substance users were presented with affectively neutral stimuli (objects used for art and craft) versus cocaine-related stimuli and had to respond on a scale from 0 (indicating “not at all”) to 10 (indicating “extremely”) to the following questions: “How good do you feel?”, “Do you have a craving or urge for cocaine?”, “Do you want cocaine?”, “Do you need cocaine?”, and “Are you turned off?” [97] (p. 12040). The authors analyzed the relationship between the change in self-reported craving and the change in metabolic brain activity across the two different sets of stimulus presentations. The results showed that an increase in craving correlates with increased metabolic brain activity in the dlPFC, amygdala, and cerebellum [97]. Moreover, one participant reporting a large increase in craving during cocaine cue presentation showed enhanced activity in the amygdala and the parahippocampal gyrus compared to a subject reporting no increase in craving during the presentation of cocaine cues [97].

Another study investigated between-group differences (using fMRI) in two groups of addicted individuals (short- and long-term abstinence) showing the activation of the hippocampus when abstinent heroin-addicted individuals were presented with heroin-related cues versus affectively neutral cues [101]. Hippocampus activity, as well as dlPFC, amygdala, and cerebellum [97,101], was reduced for long-term abstinence compared to short-term abstinence [101]. The authors suggested that long periods of abstinence can decrease the salience of environmental drug-related cues and the activity of brain regions associated with memory processes, possibly reducing the risk of relapses [101]. The increased activation of the hippocampus/parahippocampal gyrus is consistent with other research studies [97,100,106,112,113].

Moreover, abstinent SUD participants exposed to videotapes/images depicting drug-related cues vs. affectively neutral scenes/images showed higher scores in self-reported craving correlating with enhanced activity in the ACC [95,102,105,108], PCC [99,114], NAc [99,100,103], the OFC [93,94,99], the vmPFC [99,105], mPFC (i.e., middle frontal gyrus) [98], the VTA [96,104,106] and, finally, the dorsal striatum [99,107]. Furthermore, the effects of early alcohol abstinence (24 h) have been investigated through resting-state EEG and cue-reactivity fMRI experiments [99]. The authors found beta-band alterations in abstinent alcohol-addicted individuals compared to healthy individuals from a normative EEG database [99]. Source localization analysis (sLORETA) revealed increased beta2 (18.5–21 Hz) activity in the dorsal ACC (dACC) and increased beta3 (21.5–30 Hz) activity in the pregenual part of the ACC (pgACC) extending to the vmPFC [99]. Historically, EEG beta activity has been associated with emotional processing [115] as well as cognitive and sensorimotor skills involved in different tasks [116]. Increased resting-state beta frequency power has been observed in other studies investigating the spontaneous neurophysiological activity after different periods of abstinence: (a) after 90 days in the context of cocaine misuse [110], (b) from 1 to 6 months of abstinence in the context of alcohol and cocaine misuse [109], (c) after 7 days of abstinence in the context of alcohol misuse [111], and (d) from 6 days to 4.5 months of abstinence in the context of heroin misuse [117].

The co-occurrence of increased resting-state beta band and BOLD signal in the prefrontal part of the brain (dACC, pgACC, and vmPFC) denotes that these areas play a critical role in encoding different aspects of substance-related craving and that the beta frequency band may represent an electrophysiological signature of craving in a resting state design [99]. Although the construct of memory has not been directly tested, these studies suggest that brain activations underlying the experience of craving may entail the recall of emotionally filled moments. Indeed, those findings denote a considerable overlap between the localization of the neural correlates of craving and the neural activations involved in processing autobiographical memories and MEAMs (Table 1).

## 4. The Role of Perineuronal Nets in Addiction Memory Neuroplasticity

As already introduced in Section 1, craving is defined by neurobiological models of SUD as the memory of drugs’ rewarding effects superimposed upon a negative emotional state [10,11]. This means that there is a composite memory related to the drug experience (i.e., addiction memory; Figure 1) as well as a negative emotional state related to the interruption or reduction of a substance that may drive an aversive state and be characterized by psycho-physiological symptoms varying across substances [2,88]. Negative emotions experienced during withdrawal and the perception of drug-associated stimuli may induce craving because of addiction memories that are acquired and consolidated when the individuals are about to consume the drug, when the drug is being consumed, and when the individual feels the effects of withdrawal from the substance. This phenomenon can be ascribed to a state-dependent learning mechanism occurring when the subject is in the same sensory context and physiological state as during the encoding stage [120] and can lead to a relapsing state [121,122,123].

In addition to theoretical frameworks linking drug craving to the activation of brain structures involved in processing autobiographical memories, MEAMs, and the emotional content of them, it is crucial to look at molecular aspects linking memory and craving to turn a spotlight on potential neuroplastic mechanisms of change. In this section, we will particularly focus on the role of specific molecular aggregates called Perineuronal nets (PNNs) in the acquisition and reconsolidation of addiction memories [24,118,119].

Neurobiological accounts of memory formation identify three different stages: acquisition, consolidation, and reconsolidation [124,125]. Acquisition is a process by which new information is attended to, encoded, and linked to existing information in memory. The quality of this process is critically relevant to the possibility and the accuracy of reconsolidation [125]. Consolidation and reconsolidation of memories refer to a stabilization process of transient memories: while consolidation stabilizes newly acquired memories, reconsolidation restabilizes reactivated (i.e., retrieved) memories (Alberini, 2005; Kandel et al., 2013). In other words, the reconsolidation process involves bringing back to mind (retrieve) different types of information that have been stored (Kandel et al., 2013). The reconsolidation process has been shown to have a dissociable cellular and molecular basis from consolidation [126,127]. For instance, it has been shown that the administration of an antisense blocking the brain-derived neurotrophic factor (BDNF), but not the transcription factor Zif268, impairs the consolidation but not reconsolidation process, whereas administering an antisense blocking Zif268, but not BDNF, impairs reconsolidation but not consolidation process [127]. Addiction memories belong to the wider group of appetitive memories, sharing similar neurobiological correlates and the possibility of being reconsolidated [13].

Traditionally, the recall of autobiographical memories has been considered a process susceptible to distortion. Indeed, the constructive process of autobiographical memory reconsolidation has been compared to the perception of elements of the environment [125]. Sensory perception is an active way of attending to and acquiring external information through afferent, bottom-up pathways conveying the information to the brain [124,125]. From a cognitive point of view, it is a constructive process “in the sense that individuals perceive the environment from the standpoint of a specific point in space as well as a specific point in their own history” [125] (p.1448). Similarly, the reconstruction of a past event may involve several cognitive strategies including comparison, inference, shrewd guessing, and supposition, to generate a transformed memory [125].

Furthermore, neurobiological studies reveal that intracellular protein degradation mechanisms make the memory susceptible to a potential change, thus supporting this plastic aspect of the reconsolidation process [128,129,130,131,132,133,134]. Additionally, recent preclinical studies have shown that PNNs have a role in addiction memory neuroplasticity and are found in the human brain regions responsible for emotional processing, autobiographical memory, MEAMs, and the sensation of craving (Table 1) [118,119,135,136].

**Figure 1 brainsci-13-00259-f001:**
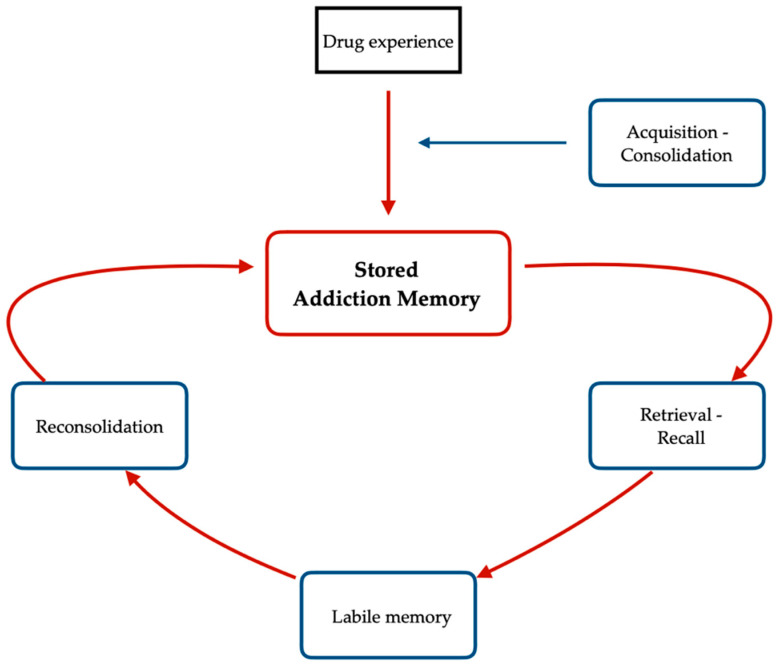
Schematic representation of the reconsolidation process of addiction memories. An experience is acquired (acquisition) and it becomes stronger over time (consolidation). When these memories are retrieved or recalled, they become labile, and this is related to specific intracellular protein degradation mechanisms—ubiquitin-proteasome system [128,130,131,133]. At this stage, memories are susceptible to being strengthened, weakened, or updated by pharmacological or non-pharmacological treatments for certain periods of time that are estimated from minutes to hours [13,133].

### 4.1. What Is a Perineuronal Net and How It Is Related to Emotional Memory

PNNs are condensed molecular aggregates of an extracellular matrix enwrapping the surface cell body, dendrites, and axon initial segments of several mammalian neuronal cells [135]. PNNs are present throughout the central nervous system and are believed to play a crucial role in brain maturation, plasticity, memory regulation, and drug addiction [135]. They have a net-like structure with holes occupied by synaptic boutons of afferent neurons that synapse on the neuron enclosed by a PNN (Figure 2). They belong to inhibitory and excitatory neurons of the brain and have been reported in several species such as mice, rats, and humans [135] representing a critical contributor to the synaptic plasticity of an emotional memory system centered in the amygdala [136].

A preclinical study [136] investigated the developmental trajectories of fear conditioning which is a historically well-studied example of emotional memory [137]. In short, the animals show a fearful response (e.g., freezing) to the delivery of a painful stimulus (e.g., a footshock) [137]. After the iterated pairing of this painful unconditioned stimulus with a neutral stimulus, which can be any general context or particular cue, the animals learn that the presence of a neutral stimulus predicts the occurrence of the painful one. This kind of associative learning transforms the neutral stimulus into a conditioned stimulus and the animal shows a fearful response when presented with the neutral stimulus. Repeated exposure of the conditioned stimulus without the aversive one can lead to extinction. That is, after appropriate training, there is a decrease in the fearful response. However, in adult animals, the fearful response can spontaneously recover if they are exposed to the original unconditioned stimulus. This is known as reinstatement and is a mechanism of the reappearance of conditioned response after extinction [138]. This phenomenon indicates that memories can resist extinction which does not erase them but involves new learning mechanisms inhibiting conditioned fear responses [138].

While adult animals show that fear-related memories are somewhat resilient to extinction, infant animals undergo memory erasure after extinction (infantile amnesia) [139,140]. The age range of the switch in the extinction phenotypes in rats has been shown to coincide with the timing of PNNs developmental formation [136]. Moreover, after the acute destruction of PNNs in the adult mice amygdala, through the injection of a specific enzyme (Chondroitin sulfate proteoglycans degrading enzyme chondroitinase ABC (ChABC)), adult mice, like the juvenile mice, exhibited a lack of fear-related memory reinstatement [136]. In other words, the destruction of PNNs in the amygdala produced an erasure-prone memory trace of the emotional (fearful) event. Importantly, the administration of the ChABC enzyme does not provoke the underlying cell to die.

The authors interpreted this finding as indicating a possible mechanism by which PNNs prevent memory erasure after extinction training by protecting these emotional memories [136]. The specific mechanism of protection is still unclear: indeed, it could be that the PNNs enwrapping amygdala neurons preserve potentiated synapses forming the memories from the depotentiation (also known as long-term potentiation reversal) normally produced by the extinction period. However, since the removal of PNNs does not prevent extinction learning, it can be that PNNs mediate network plasticity in different connected brain areas—such as the amygdala, the prefrontal cortex, and the hippocampus—involved in information storage [136]. The functional consequence of this is that PNNs, winding around the neurons, prevent the disruption of memory reconsolidation by reducing the ability of new experiences to shape previously acquired relevant information.

### 4.2. Acquisition and Reconsolidation of Addiction Memories

As we discussed in the previous section, the role of PNNs in the synaptic plasticity of memories of experiences that evoke emotional reactions (emotional memories) and associative learning [136] justifies its investigation into drug addiction and drug-related memories. Changes in the PNNs during the consumption of drugs of abuse (heroin, alcohol, cocaine, and nicotine) have been widely documented [24]. For instance, it has been shown how acute cocaine exposure (cocaine Intraperitoneal injection of 1 mL/kg for 1 day) decreased PNNs intensity, while repeated cocaine exposure (cocaine intraperitoneal injection of 1 mL/kg for 5 days) increased PNNs intensity around the neurons of the prefrontal cortex [141], possibly “trapping” cocaine-associated memories and not allowing new information to shape it [24] (p. 197). In other words, cocaine exposure alters the density of PNNs enwrapping neurons of the prefrontal cortex which is critical to addiction memory. The interpretation of this finding is that a novel cocaine exposure allows for an increasing number of synaptic connections by decreasing the intensity of PNNs around prefrontal cortical neurons, and this may possibly help to form new state-dependent and drug-related memories. In contrast, repeated cocaine exposure is thought to determine a decreasing number of synaptic connections by increasing the thickness of PNNs around prefrontal cortical neurons, and this might impede new memories to form, locking in state-dependent and drug-related memories.

Furthermore, there have been a few studies investigating the role of PNNs in the acquisition and reconsolidation of drug-related memories in the animal model [118,119]. One of the main behavioral paradigms used in preclinical studies to investigate different aspects of drug addiction is the conditioned place preference test (CPP). It is assumed that animals can learn to associate a particular environment with drug treatment and a different environment with the absence of it. The apparatus may contain a varying number of compartments that can be discriminated by the animals and that will be associated with different experimental conditions. During the training phase, the animal is injected with a drug that can have potentially rewarding or aversive effects and is placed into one of the compartments of the apparatus for several minutes. On the following day, the animal is injected with a saline substance with no physiological effects and placed in a different compartment. The compartments associated with the different experimental conditions have strictly different physical properties. For instance, one of them may have black walls with a wire mesh floor and the other one white walls with a metal rod floor.

In preclinical research studies that we will describe below [118,119], the animals have been trained to pair one side of the box with cocaine and the other with a saline substance [118,119]. The alternation of the drug with the saline control condition can last a varying number of days. After the training phase, a test session is conducted to measure the time the animal spends in each compartment. The conditioned place preference occurs when the animals spend a significantly greater amount of time in the drug-paired compartment compared to the saline-paired compartment, and this generally happens with drugs of abuse such as cocaine [142]. On the other hand, in case the animals spent a greater amount of time in the saline-paired compartments versus the drug-paired compartments, we would have a conditioned place aversion behavior (CPA) that happens, for instance with lithium chloride [142]. CPP is generally associated with two procedures: extinction and reinstatement. During the extinction training, the animals are allowed free access to all compartments and no drug is delivered. In this way, the conditioned stimulus is presented repeatedly without the presence of the unconditioned one (i.e., the drug, in this case) so that the association between the compartment and the aversive or rewarding stimulus is reduced [142]. After the extinction procedure, a reinstatement induction is often used to translate in the animal model a human relapsing state. Reinstatement occurs when the animals’ behavior reacquires a CPP after the extinction training [142]. The way to induce a reinstatement is generally two-fold: by administering a lower dose of a drug or by providing a stressful stimulus [142].

**Figure 2 brainsci-13-00259-f002:**
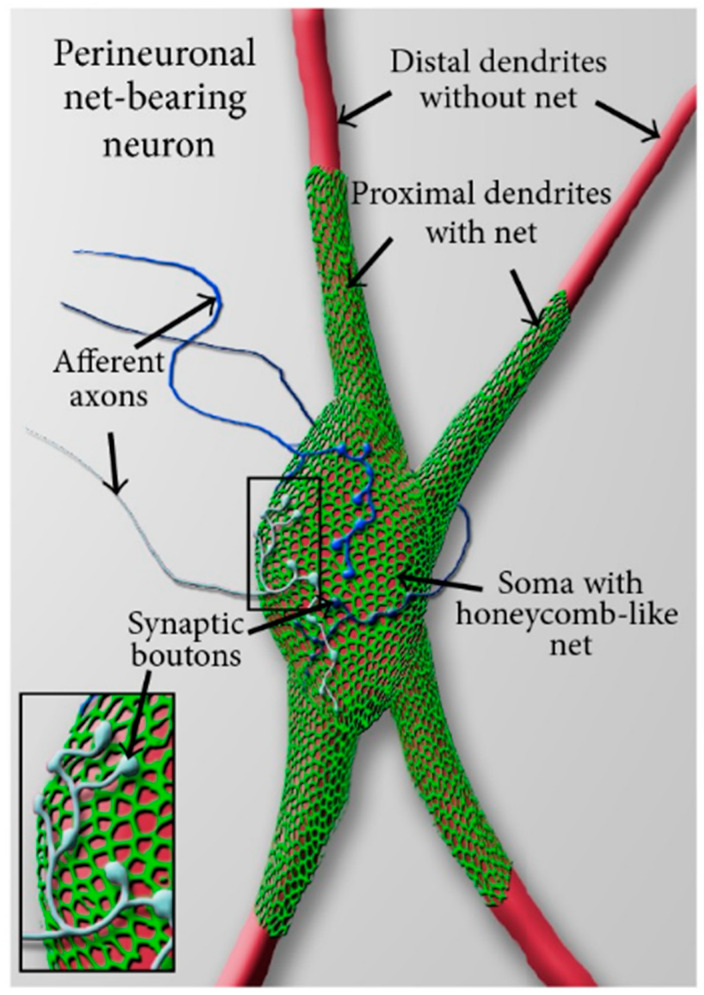
Adapted, with permission, from [143]. Schematic view of a perineuronal net-bearing neuron. A neuronal cell body (soma) with its proximal dendrites (red) covered by a typical reticular, honeycomb—like net (green). More distally, dendrites are devoid of nets. The holes in the perineuronal nets are occupied by synaptic boutons of afferent axons (insert, blue) that synapse on the net-bearing neuron.

Two recent works investigated the role of PNNs within the prelimbic mPFC (animals’ prelimbic mPFC corresponds to the human dlPFC [6]) and amygdala on the acquisition and reconsolidation (maintenance) of cocaine-induced memories [118,119]. In animals injected with the ChABC in the prelimbic mPFC and in the basolateral nucleus of the amygdala before the CPP, the development of cocaine-conditioned place preference is attenuated. This suggests that the memory relative to the association between the environment in which they experienced the drug and the internal rewarding effect (achieved while on the drug) is disrupted with the removal of PNNs [118,119]. Moreover, when the destruction of PNNs, caused by ChABC, is tested after the extinction of the cocaine-conditioning but before a reactivation session, the cocaine-conditioned place preference is attenuated again suggesting a critical role of PNNs in the acquisition but also in the reconsolidation of drug-related memories [118,119].

These results are in accordance with previous studies [136] demonstrating the effect of intra-amygdala PNNs removal by ChABC injection on the reinstatement of emotional memories (fear-related) and offer a supporting link to neuroimaging findings in humans (see Section 3.1) highlighting the role of a memory circuit including the dlPFC and the amygdala during the experience of craving. In conclusion, the causal role of PNNs in the acquisition of state-dependent and drug-related memories and in the reconsolidation process indicates cortical and subcortical plastic mechanisms underlying the state-dependent recalls that can possibly be the target of addiction treatment.

## 5. Conclusions

Neuroimaging studies of drug cravings showed changes in areas of the brain that are critical to the reconsolidation process of autobiographical memories suggesting that the recall of emotionally filled moments may underlie the experience of craving. Here, we show a remarkable overlap between these regions of the brain and the areas involved in MEAMs that are generally characterized by emotional content (Table 1). Those similar activations are summed up in the following paragraph.

The dlPFC has been postulated to encode the search and controlled processes of autobiographical memory retrieval [66,67,69], and its increased metabolic activity positively correlates with MEAMs [82,83,84] and with the feeling of craving [97,101]. The vmPFC has been associated with a pre-conscious form of monitoring during autobiographical memory retrieval [66,67,76], with the retrieval of autobiographical memories while listening to familiar and popular songs [82], especially for younger adults [83], and with a craving state [99,105]. The activation of mPFC and PCC has been associated with self-referential processes during autobiographical memory retrieval [66,67,69,77], with the retrieval of MEAMs [82,83,84], and with craving state: (mPFC activation in [98] and PCC activation in [99,114]). The OFC, the amygdala, and the hippocampus have been associated with the processing of emotional content and vivid sensory details of autobiographical memories [38,65,67,69,70], with the retrieval of MEAMs [82,83] and with higher scores in self-reported craving (OFC activation in [93,94,99], amygdala activation in [97,101], and hippocampus activation in [101,106,112,113]). These changes in functionally relevant brain structures may help to explain the compulsive and relapsing nature of SUD [89] and why craving is more likely to occur in an environment where the drug was obtained or used and, thus, a memory is formed [2].

It has been proposed that music can be a trigger for craving through a process of state-dependent recall [15] which is part of the state-dependent learning theory [120] if particular elements or segments of the music remind the user of the intensity of drug action and music enforcing each other. Furthermore, a downward spiral [17] has been described, in which an initial phase of ‘falling in love’ with the drug-enhanced enjoyment of music is followed by a ‘downhill’ phase in which music, but not the drug, becomes less important, ending with indifference towards the music. Or as Eric Clapton described it: “Part of the trap [of drugs and alcohol] is that they open the doors to unreleased channels or rooms you hadn’t explored before or allowed to be open…Unfortunately after that, the booze becomes more important than the doors it’s opening, so that’s the trap” [144] (p.199). Thus, memories of the first stage may especially trigger a craving. Consequently, it has been suggested that MT and MBIs interventions may induce a craving state in SUD clients [45,57,58], and a recent systematic review’s main finding suggests that music can induce substance craving in non-therapeutic settings, providing a further contribution to previous conceptualizations and regarding good MT/MBIs clinical practice [145].

However, this risk can be handled within a music therapy setting, transforming, and retraining the emotional components related to addiction memories and reframing the experience of the ‘dangerous music’ and its elements cueing drug memories [16,17,45,59]. While music can heighten emotions and cue recalls, the music therapist can promote awareness of the connection between music and emotions making this stimulus susceptible to revaluation [17]. Further to this, the complex communicative channel that a music therapist can develop with the client may help to reframe the narrowed focus of attention onto alternative opportunities of reward processing and recalibrate the emotional intensity of the music experience in the therapy setting by deconditioning memory content and musical valence [15]. In the present article, starting from the observation of remarkable similarities in the activity of brain regions involved in autobiographical memory retrieval, MEAMs, and the feeling of craving, we propose the neuroplastic feature of the reconsolidation process of addiction memories—demonstrated through the measurement of PNNs in the dlPFC and amygdala—as an explanatory viable angle to update and recalibrate the emotional content related to the recall of addiction memories.

Although the phenomenological characteristics of addiction memories are still elusive, a recent study has validated the “Addiction Memory Intensity Scale” showing a promising tool to measure addiction memories in clinical research [146]. It has been suggested that the characteristics of autobiographical memories can be taken as a frame of reference to study addiction memories because they are both related to personal experiences that, in the case of addiction memories in humans, are related to the individual’s history of drug use [146]. The recall of autobiographical memories is a labile process [125] and the reconsolidation stage depends on specific intracellular protein degradation mechanisms (the ubiquitin/proteasome—dependent protein degradation pathways) that make the memory susceptible to a possible modification [128,129,130,131,132,133,134]. Authors suggested that, at the reconsolidation stage, those memories can be strengthened, weakened, or updated by pharmacological or non-pharmacological treatments for a period of time that is estimated from minutes to hours [13,133]. Several studies conducted on the animal model suggest that it is possible to disrupt the reconsolidation of drug-associated memories by manipulating cellular and molecular signaling pathways in the nucleus accumbens, dorsal striatum, and amygdala by pharmacological agents [13]. Indeed, further to pharmacological preclinical studies intervening in drug memory reconsolidation, there is evidence of non-pharmacological manipulations disrupting memory reconsolidation in drug addiction [147]. For instance, a preclinical study suggested how sleep deprivation may disrupt the reconsolidation of memories for the reinforcing effects of drugs (in this case, morphine) in addicted rats [147]. This neuroplastic feature linked to the recall of general autobiographical memories is further supported by research into the activity of PNNs. The PNNs in the dlPFC and the amygdala causally regulate the acquisition and the reconsolidation of addiction memories indicating cortical and subcortical plastic events underlying the recall of addiction memories and the experience of craving [118,119].

The research on PNNs represents great evidence of the dynamic and changeable characteristic of addiction memory that can be possibly attenuated or updated to suppress the cycle of relapse to drug use [13,133]. Pharmacological and non-pharmacological therapies may target PNNs and possibly influence the reconsolidation of addiction memories. Interestingly, PNNs’ presence throughout the brain is reduced in those animals exposed to enriched environments, possibly boosting plasticity and improving learning performance [118,119,148,149]. Thus, the fact that PNNs are highly sensitive to environmental stimulation [148,149] makes it reasonable to develop appropriate non-pharmacological treatments to alter brain plasticity imposed by drugs of abuse.

### 5.1. Implications for Music Therapy Research and Clinical Practice

The neuroplastic characteristics of addiction memory may serve as a basis for updating and/or recalibrating the emotional content associated with the recall in MT/MBIs settings. With this article, we support the development of long-term MT and MBIs interventions where the therapist can engage those plastic mechanisms and positively influence addiction memories through the use of music. MT and MBI settings might facilitate MEAMs that are possibly linked to the drug experience. However, within this safe environment, with the support of the therapist, music may represent a tool to evoke new sensations, moods, and emotions [149,150] during tense moments due to the recall of addiction memories.

Here, it is proposed that the emotional content attached to addiction memories can be updated and/or recalibrated via the ability of music to evoke emotions within a therapeutic setting where the therapist supports the client to process these moments. Furthermore, recent works have been analyzing the affiliative bonding between client and therapist in music-based and non-music-based therapeutic settings by measuring interpersonal brain synchronization [151,152]. This presents promising opportunities to study the temporal dynamics of event-related peaks of emotional processing in MT and MBIs approaches for SUD. Those advances may provide an index of the therapeutic change and an ecological measure of the therapeutic relationship.

Finally, it is in line with influential models of SUD suggesting the need to investigate a personalized clinical approach instead of the standard diagnostic-based view (i.e., the equifinality concept of addictive behavior in [4]). We advocate for the use of these advanced methods (EEG hyperscanning suggested in the context of MT [151] and fNIRS suggested in the context of psychological counseling [152]) to study the formation of new state-dependent memories in MT/MBIs settings, provide potential indices of therapeutic change, and disseminate useful knowledge to stimulate future clinical trials [46] and clinical practice.

### 5.2. Strengths and Limitations

To the best of our knowledge, this is the first narrative review that aims to create a common thread between different scientific domains, such as MT/MBIs for SUD, the neuroplasticity of addiction memories, and the neuroscience of drug craving, while posing the question of whether MT and MBIs are useful therapeutic tools for treating addiction memories in SUD. Additionally, it valued evidence from a variety of scientific approaches, integrating them to reach a novel conclusion. A first limitation is that, for the reasons listed in the Methods subsection, we decided to conduct a narrative-style scientific review, subjecting the current article to limitations such as potential bias in article selection and robustness in the methodological approach.

However, using a narrative review instead of a systematic review allowed us to integrate diverse methodological and theoretical backgrounds that are crucial for a field of research that is so interdisciplinary to be constrained from restrictions based on study design or outcome measures. It is also important to note that SUD and the experience of craving are broad topics encompassing social and cultural levels of analysis that are not part of the approach we adopted for this article. Thus, we decided to confine the theoretical framework behind our constructs of interest to the description of neurobiological and neurocognitive models at the cost of reducing the complexity of their narrative account.

## Figures and Tables

**Table 1 brainsci-13-00259-t001:** Evidence for the overlap between brain structures involved in the feeling of craving, autobiographical memory retrieval, and MEAMs.

Brain Region	BA	Domain Involved	Main Findings
dlPFC	46	AM	↑ activation during AM recollection [66]
	46	MEAM	↑ activation during familiar and popular music triggering AM [82,84]
	46	Craving	↑ activation during the presentation of drug-related cues [101]
	N.A. *	PNNs	Causal role in acquisition and reconsolidation of addiction memories in preclinical study [118]
vmPFC	9/10	AM	↑ activation during AM recollection [76]
	25/11	MEAM	↑ activation during familiar and popular music triggering AM [82]
	11/32/25	Craving	↑ activation during the presentation of drug-related cues [105]
mPFC	10/32	AM	↑ activation during AM recollection [66]
	8/9/10/11/32	MEAM	↑ activation during familiar and popular music triggering AM [82,84]
	9/10	Craving	↑ activation during the presentation of drug-related cues [98]
OFC	11	AM	↑ activation during emotional content recollection [65]
	11	MEAM	↑ activation during familiar and popular music triggering AM [82]
	11/47	Craving	↑ activation during the presentation of drug-related cues [93,94]
PCC	23/29/30/31	AM	↑ activation during AM recollection [66,69]
	30/31	MEAM	↑ activation during familiar and popular music triggering AM [82]
	31	Craving	↑ activation during the presentation of drug-related cues [114]
Amygdala	N.A.	AM	↑ activation during emotional content recollection [65,69]
	N.A.	MEAM	↑ activation during familiar and popular music triggering AM [82]
	N.A.	Craving	↑ activation during the presentation of drug-related cues [97,101]
	N.A. *	PNNs	Causal role in acquisition and reconsolidation of addiction memories in preclinical study [119]
Hippocampus	N.A.	AM	↑ activation during AM recollection [66]
	N.A.	MEAM	↑ activation during familiar and popular music triggering AM [82]
	N.A. *	Craving	↑ activation during the presentation of drug-related cues [97,100,101,106,112,113]

Note: * Information not available because it is a preclinical study. Arrows indicate significant greater activity relative to control conditions. Abbreviations: BA, Brodmann Area; dlPFC, dorsolateral prefrontal cortex; vmPFC, ventromedial prefrontal cortex; mPFC, medial prefrontal cortex; OFC, orbitofrontal cortex; PCC, posterior cingulate cortex; AM, autobiographical memory; MEAM, music-evoked autobiographical memory; PNNs, perineuronal nets; N.A. not available.

## Data Availability

No data were used to support this study.

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
