# Peer review of "Neuroplastic Changes in Addiction Memory—How Music Therapy and Music-Based Intervention May Reduce Craving: A Narrative Review"

_brainsci, 2023, doi:10.3390/brainsci13020259_

Round 1
Reviewer 1 Report
Thank you for this very interesting and thought-provoking manuscript. I found it to be very well written, and I believe it could lead to important clinical applications. I just have a few comments/suggestions/questions.
1. Please look at the paragraphs within the manuscript as it is best practice to have at least 3 sentences per paragraph. There are several times that paragraphs are made up of only one sentence, and these could easily be combined with the previous or the following paragraph.
2. Line 39 - it might make more sense (and read better) if "objectifyingly" and "insultingly" were changed to "objectifying" and "insulting".
3. Lines 60-62 - one of the most important aspects of music therapy which was left out of the definition provided here is the therapeutic relationship. This is especially important to include since section 5.1 points to the importance of the therapeutic relationship and what can be accomplished when the music therapist works with and processes with the client.
4. Please make sure your references are listed consistently and accurately. For example, pay attention to whether or not the names of journals and the titles of books should be capitalized.
Author Response
- Please look at the paragraphs within the manuscript as it is best practice to have at least 3 sentences per paragraph. There are several times that paragraphs are made up of only one sentence, and these could easily be combined with the previous or the following paragraph.
Thank you for your comments and your suggestion! Indeed, there were several paragraphs made of less than 3 sentences. We’re confident this point has been properly addressed.
- Line 39 - it might make more sense (and read better) if "objectifyingly" and "insultingly" were changed to "objectifying" and "insulting".
Thank you for this comment, we agree with you that it was not the best way to formulate the concept in line 40. I (Fi.Pa.) just wanted to cite exactly McWilliams’ words in her book McWilliams N. Psychoanalytic diagnosis: understanding personality structure in clinical process. 2nd ed.; New York: Guilford Press; 2011; p. 26. We have accepted your suggestion and it has now been corrected.
- Lines 60-62 - one of the most important aspects of music therapy which was left out of the definition provided here is the therapeutic relationship. This is especially important to include since section 5.1 points to the importance of the therapeutic relationship and what can be accomplished when the music therapist works with and processes with the client.
Thank you for your comment and your observation. We have addressed this one by adding a couple more sentences in lines 86-91 (previously lines 60-62). We hope that this point has now been properly addressed.
- Please make sure your references are listed consistently and accurately. For example, pay attention to whether or not the names of journals and the titles of books should be capitalized.
Thank you for this observation! Indeed, the reference section was inaccurate. We’ve corrected all formatting mistakes.
Reviewer 2 Report
This narrative review addresses the mechanisms and neural substrates underlying music-induced and craving-related emotions and memories with the ultimate goal of providing a foundation for music-based therapies for craving.
This is an interesting topic with a large spectrum of potential applications in empirical studies and, ultimately in therapies for drug craving. The authors tried to keep the sections organized. The literature is relevant.
My main concern relates to the way ideas are presented, which I do not think is the best one. The first paragraphs of the text (abstract included) lack general, concise statements that would help readers focus on the big picture.
Abstract
The abstract is not easy to read, mostly because the objective of the review is not clearly stated (three topics are presented, but in a fragmented, non-unitary way, unrelated to a goal). I guess the goal is to investigate how and why music-based interventions may reduce craving (it is in the title), but it would help seeing this in the abstract.
In addition, several non-trivial concepts are introduced here without any explanation (e.g., state-dependent vs. drug-related memories).
All this made me lose interest in the paper, even though, when I moved on to the introduction, it was clearer.
1-Introduction
-Before introducing the question (72-74), it would help putting some focus on the concept of neuroplasticity and how it applies to addiction memory in the context of the present paper
-87-88: I would not make a paragraph because ln 88 is an explanation of the previous sentence.
- The three-folded purpose as presented here (Music cues autobiographical memories, addiction memory engages autobiographical emotional memory, PNNs are involved autobiographical emotional memory in craving) a presented here does not provide an immediate suggestion that music may help addressing cravings. It would be helpful to articulate this with the larger purpose. Though lns 91-94 try to do this, my feeling is that the reader gets lost even so. One possibility could be moving up the contents of 91-94, before the three-folded purpose is presented.
Further in lns 98-99 there seems to be a concise and simple framing of the review’s purpose. Maybe this kind of statement should have priority in the text.
2- Music cues emotional autobiographical memories
-219-220: I would make no paragraph
-226-260: I would merge these into a single paragraph since it deals with sub-systems
-261-268: these deal with the role of emotion, which is a different topic. I would merge this.
GENERAL COMMENT: I would suggest that the authors gather all ideas that concern one big topic – only one - into one paragraph. Otherwise, chunking and relating relatable information when reading becomes very difficult.
3-craving
-Here, the take-home message is the overlap of circuits for music-induced autobiographical memories and emotions with those underlying cravings. Maybe it could be useful to anticipate this take-home message (the table is quite useful, by the way) and move it up?
GENERAL COMMENT: My feeling is that the authors proceed in a strict analysis-then-synthesis mode of presentation, which is not always useful. Sometimes, readers need to see the big picture first in order to organize what comes next.
GENERAL COMMENT: I would keep references to brain areas restricted to those that are consequent, i.e., those that help understand why the circuits overlap or some other important message related to the goal. I did not have the time to analyse in detail if the authors have already done that, but, if they did not, I would recommend it.
4-PNNs
The general comments outlined above apply here.
5-Conclusion
-Lines 610-627: has this information not been provided before? Is it necessary to restate it?
Author Response
Authors point-by-point response is in italics.
This narrative review addresses the mechanisms and neural substrates underlying music-induced and craving-related emotions and memories with the ultimate goal of providing a foundation for music-based therapies for craving.
This is an interesting topic with a large spectrum of potential applications in empirical studies and, ultimately in therapies for drug craving. The authors tried to keep the sections organized. The literature is relevant.
My main concern relates to the way ideas are presented, which I do not think is the best one. The first paragraphs of the text (abstract included) lack general, concise statements that would help readers focus on the big picture.
Thank you for your comments and your helpful suggestions.
Abstract.
The abstract is not easy to read, mostly because the objective of the review is not clearly stated (three topics are presented, but in a fragmented, non-unitary way, unrelated to a goal). I guess the goal is to investigate how and why music-based interventions may reduce craving (it is in the title), but it would help seeing this in the abstract.
The abstract has been rephrased in light of reviewer comments. The goal of the present work has been clearly stated in line 19.
In addition, several non-trivial concepts are introduced here without any explanation (e.g., state-dependent vs. drug-related memories).
Thank you for pointing out a potential misunderstanding that we tried to address in the revision. However, the abstract cannot be the place to explain complex concepts such as state-dependent memories that are not presented as contrasting (as you indicated with “versus”) with drug-related memories. On the contrary, both elements (i.e., state-dependent and drug-related memories) are part of addiction memories. However, drug-related and music memories are connected during state-dependent recall of their combined emotional magnitude, resulting in a specific drug experience linked to music that can also induce craving. We further added that it is this seeming contradiction that we want to address in this narrative review and suggest a solution based on the causal role of PNNs in addiction memory neuroplasticity.
All this made me lose interest in the paper, even though, when I moved on to the introduction, it was clearer.
We are sorry for this, and hopefully the abstract and the rest of the paper are clearer now. Thanks again for your comments and corrections.
1-Introduction
-Before introducing the question (72-74), it would help putting some focus on the concept of neuroplasticity and how it applies to addiction memory in the context of the present paper
Thank you, a sentence about neuroplasticity in relation to addiction memory has been added in lines 78-82. This hopefully helped clarity.
-87-88: I would not make a paragraph because ln 88 is an explanation of the previous sentence.
Thank you for the observation. It has now been corrected.
- The three-folded purpose as presented here (Music cues autobiographical memories, addiction memory engages autobiographical emotional memory, PNNs are involved autobiographical emotional memory in craving) a presented here does not provide an immediate suggestion that music may help addressing cravings. It would be helpful to articulate this with the larger purpose. Though lns 91-94 try to do this, my feeling is that the reader gets lost even so. One possibility could be moving up the contents of 91-94, before the three-folded purpose is presented.
Further in lns 98-99 there seems to be a concise and simple framing of the review’s purpose. Maybe this kind of statement should have priority in the text
Thank you for the suggestion on how to present the three-folded purpose. We have decided not to move up the contents of 91-94 (now at the end of section one, lines 127-132) but to broaden the question preceding the three folded purpose description in order to provide a clearer picture of our aim and of the final idea we will present. We also modified accordingly the lines following the question, including giving more relevance to lines 98-99 (now at lines 131-132).
2- music cues emotional autobiographical memories.
-219-220: I would make no paragraph
-226-260: I would merge these into a single paragraph since it deals with sub-systems
-261-268: these deal with the role of emotion, which is a different topic. I would merge this.
GENERAL COMMENT: I would suggest that the authors gather all ideas that concern one big topic – only one - into one paragraph. Otherwise, chunking and relating relatable information when reading becomes very difficult.
Lines 219-220 (now, lines 255-256) have been corrected, thanks for noticing
Lines 226-260 (now, lines 262-294) have been merged, thanks.
3- craving
-Here, the take-home message is the overlap of circuits for music-induced autobiographical memories and emotions with those underlying cravings. Maybe it could be useful to anticipate this take-home message (the table is quite useful, by the way) and move it up?
GENERAL COMMENT: My feeling is that the authors proceed in a strict analysis-then-synthesis mode of presentation, which is not always useful. Sometimes, readers need to see the big picture first in order to organize what comes next.
GENERAL COMMENT: I would keep references to brain areas restricted to those that are consequent, i.e., those that help understand why the circuits overlap or some other important message related to the goal. I did not have the time to analyse in detail if the authors have already done that, but, if they did not, I would recommend it.
Thank you for the first “general comment”, we’ve added a sentence at the beginning of section 3 describing the scope of the section.
Thank you for the second “general comment”, we have re-shaped some parts of the article and added more sentences to explicitly state/improve the communication regarding the scope of the single sections and, in general, of the entire article.
3- craving and 4- PNNs
The description of the neurophysiological activity examined in the papers analyzed here, has been done including only those activations/ those results that were overlapping across our domains of interest.
5- conclusion
-Lines 610-627: has this information not been provided before? Is it necessary to restate it?
Yes, you’re right that information is already provided before in different sections but in conclusion, we wanted to gather the similar brain activations in a sum-up paragraph.
Round 2
Reviewer 2 Report
My response is highlighted in bold in the attached file.

Author Response
Dear Reviewer,
thank you for your comments and suggestions. They helped us improve our article, and, especially the abstract of it.
We have followed your suggestions and rewrote the abstract according to them!
We think that the abstract's new shape makes it clearer, more logical and, more linear, making it easier (for laypersons as well) to understand the language.
Thank you again for your comments and kind regards.